# DeeDiff: Dynamic Uncertainty-Aware Early Exiting for Accelerating Diffusion Model Generation

## Abstract

Diffusion models achieve great success in generating diverse and high-fidelity images. The performance improvements come with low generation speed per image, which hinders the application diffusion models in real-time scenarios. The generation of diffusion models generally requires multi-step network inference. While some certain predictions benefit from the full computation of the model in each sampling iteration, not every iteration requires the same amount of computation, potentially leading to computation waste. In this work, we propose DeeDiff, an early exiting framework that adaptively allocates computation resources in each sampling step to improve the generation efficiency of diffusion models. Specifically, we introduce a timestep-aware uncertainty estimation module (UEM) for diffusion models which is attached to each intermediate layer to estimate the prediction uncertainty of each layer. The uncertainty is regarded as the signal to decide if the inference terminates. Moreover, we propose uncertainty-aware layer-wise loss to fill the performance gap between full models and early-exited models. With such loss strategy, our model is able to obtain comparable results as full-layer models. Extensive experiments of class-conditional, unconditional, and text-guided generation on several datasets show that our method achieves state-of-the-art performance and efficiency compared to existing early exiting methods on diffusion models. More importantly, our method even brings extra benefits to baseline models and obtains better performance on CIFAR-10 and Celeb-A datasets. Full code and model are released for reproduction.[1]

## 1 Introduction

Diffusion models have shown significant performance improvement in generating diverse and high-fidelity images. A line of recent works Ho et al. (2020); Balaji et al. (2022); Song et al. (2020a); Kim et al. (2021); Li et al. (2022b); Lu et al. (2022a); Sehwag et al. (2022); Austin et al. (2021) demonstrate the superiority compared with state-of-the-art GAN Brock et al. (2018); Lee et al. (2021); Goodfellow et al. (2020) models on different tasks such as unconditional image generation Ho et al. (2020), image editing Li et al. (2022a), video generation Ho et al. (2022) and text-guided image generation Ramesh et al. (2022). However, one drawback of diffusion models is the large computation resources requirement for progressive generation, which hinders the wide application of diffusion models. Specifically, the generation process of the diffusion model requires multi-step noise estimation where each step represents a single computation of the neural network. Such a procedure leads to a low-speed generation of diffusion models compared to GAN models.

The speed of image generation within diffusion models is primarily determined by two factors: the number of generation steps and the latency of model inference at each step. In attempts to enhance generation efficiency, previous approaches Song et al. (2020a); Lu et al. (2022b;c) have predominantly focused on curtailing the number of sampling steps. Nonetheless, these methods still involve employing the same amount of network computation for each individual sampling step, therefore leading to suboptimal generation speed. Consequently, a natural question arises: *is it necessary to use full models with all layers for all timesteps in diffusion generation?*

---

[1]Project Repo: `https://anonymous.4open.science/r/DeeDiff-E0F7/`

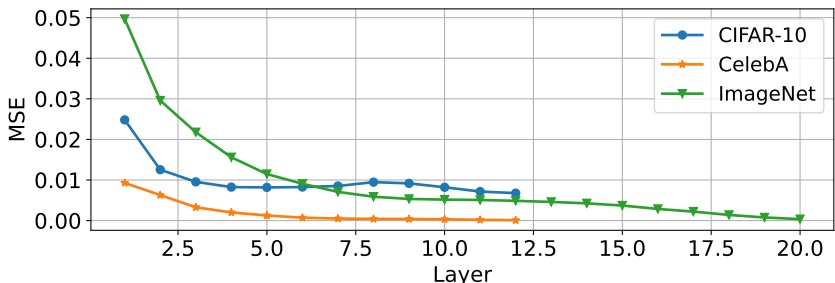

Figure 1: Average MSE loss between intermediate layer outputs and final layer output across CIFAR-10, CelebA, and ImageNet datasets.

In this paper, we assert the existence of redundancy within the backbone networks utilized for multi-step generations in diffusion models. The evidence is shown in Figure 1. The figure illustrates the trend of average Mean Square Loss (MSE) between each layer's intermediate output and the final layer's output across generation steps on CIFAR-10 Krizhevsky et al. (2009), CelebA Liu et al. (2015), and ImageNet Deng et al. (2009). On CIFAR-10 and CelebA datasets, the backbone is the Transformer structure consists of 13 layers while on ImageNet, the backbone includes 21 layers. The loss drops sharply in the initial layers and then reaches a plateau in the middle layers across all datasets, indicating that the intermediate representations stabilize before the final layers. This suggests shallow layers can potentially generate results comparable to the deep layers, motivating the possibility of early exiting approaches.

Early exiting Teerapittayanon et al. (2016) aims to reduce network redundancy during inference by allocating computation resources adaptively in terms of input complexity. While early exiting has shown promise in classification and text generation by leveraging class-level or token-level confidence Xin et al. (2021); Schuster et al. (2022), applying it to diffusion models for image synthesis poses challenges. First, unlike text where each token provides uncertainty to guide early exiting, the iterative generation process in the diffusion model lacks natural confidence estimation. Second, the generation steps of diffusion models represent different stages in progressively denoising and constructing the final image, rather than textual tokens. Thus, errors accumulate across the multi-stage generation process Hang et al. (2023), posing additional challenges for early exiting compared to text generation. Overall, existing confidence-based early exiting techniques developed for classification and textual token generation are insufficient for the multi-stage image generation process of diffusion models. New approaches are needed to enable efficient inference while retaining the complex image synthesis capabilities.

To address these challenges, we propose DeeDiff, a novel early exiting approach tailored to diffusion models. Unlike prior methods relying on classification confidence, DeeDiff introduces a sampling uncertainty estimation module (UEM) to guide exits based on sampling uncertainty at each layer. A timestep-aware design accounts for the distinct stages of the sampling process. Further, a new uncertainty-weighted, layer-wise loss uses the estimated uncertainties to reshape losses during training to recover performance lost from early exiting. Experiments demonstrate that DeeDiff is able to reduce inference time with a minimal performance drop compared to other SoTA methods. Moreover, our method boosts the performance of models, showing better FID on CIFAR-10 and Celeb-A datasets when early exiting is not applied. Our main contributions are summarized as follows:

- To the best of our knowledge, this is a pioneering work to extend early exiting to accelerate the inference of diffusion models. To this end, we propose a novel early exiting framework called DeeDiff with a valid assumption on different computation requirements of different sampling steps in diffusion models. It is worth noting that our DeeDiff framework can be easily plugged into any current CNN-based and Transformer-based diffusion model.

- We introduce a timestep-aware uncertainty estimation module (UEM) in diffusion models to estimate the uncertainty of each prediction result from intermediate layers at each sampling step. The uncertainty estimated by UEM is utilized as an exiting signal to decide the timing of exiting.

- In order to fill the performance gap between full models and early-exited models, we propose an uncertainty-aware layer-wise loss that reweights the layer-wise loss with estimated uncertainty from the uncertainty estimation module (UEM). The experiments show that the loss strategy brings extra benefits to our model and achieves better performance to full-layer models without early exiting.

- Extensive experiments on unconditional and class-conditional generation and text-guided generation show that our method can largely reduce the inference time of diffusion models by up to 40% with minimal performance drop. On CIFAR-10 and Celeb-A, our model even brings extra benefits on top of the baseline model and obtains better performance without exiting.

## 2 RELATED WORK

**Denoising Diffusion Models.** With the strong ability to generate high-fidelity images, diffusion model Ho et al. (2020); Dhariwal & Nichol (2021); Song & Ermon (2019); Song et al. (2020b) achieved great success in many applications such as unconditional generation Ho et al. (2020); Song et al. (2020b); Karras et al. (2022), text-guided generation Balaji et al. (2022); Rombach et al. (2022); Saharia et al. (2022); Ramesh et al. (2022); Chefer et al. (2023), image inpainting Lugmayr et al. (2022) and so on. Diffusion models are superior in modeling complex data distribution and stabilizing training in comparison with previous GAN-based models Brock et al. (2018); Xu et al. (2018); Zhou et al. (2021); Zhu et al. (2019), which suffer from unstable training. Ho et al. (2020) first proposed to utilize a neural network to estimate noise distribution. The backbone structure used in theirs and most other diffusion models is UNet Ronneberger et al. (2015). More recently, a line of works Bao et al. (2022a); Peebles & Xie (2022) have also explored the application of Transformers Dosovitskiy et al. (2020) as a backbone network, with U-ViT Bao et al. (2022a) utilized the long skip connection and leveraged Adaptive LayerNorm to achieve SoTA performance on image generation.

**Efficiency in Diffusion.** One of the drawbacks of diffusion models is the low generation speed. On the one hand, diffusion models require multi-step (eg. 1000 steps) gradual sampling to generate high-fidelity images. On the other hand, the computation resource of one step is also expensive. Most existing works Song et al. (2020a); Luhman & Luhman (2021); Salimans & Ho (2022); Meng et al. (2022); Lu et al. (2022b;c); Lyu et al. (2022); San-Roman et al. (2021); Watson et al. (2022) target on reducing sampling steps. Specifically, Luhman & Luhman (2021); Salimans & Ho (2022); Meng et al. (2022) utilize the knowledge distillation method to insert multi-step teacher knowledge into the student model with fewer sampling steps. The drawback of these methods is they require large computation resources for distillation training. In contrast, Song et al. (2020a); Lu et al. (2022b;c); Jolicoeur-Martineau et al. (2021); Bao et al. (2022b) are able to reduce sampling steps to 50 with minimal performance loss, and no re-training is required. Besides on reducing sampling steps, Li et al. (2023) applies quantization to diffusion backbone to reduce GPU memory and increase backbone inference speed. Lyu et al. (2022) applied early stop during training to accelerate diffusion. However, all these methods utilize the full amount of network computation for each sampling step. Moon et al. (2023) proposed a static early exiting method to accelerate the inference of diffusion models. Such a predefined early exiting strategy achieves suboptimal performance and efficiency trade-off. Instead, Our method aims to dynamically allocate computation for inputs from different sampling steps, which brings benefits for the reverse denoising process.

**Early Exiting Strategy.** Early exiting Teerapittayanon et al. (2016) is a kind of method for neural network acceleration. The main assumption of early exiting is that different inputs require different computation resources. Existing early exiting methods Teerapittayanon et al. (2016); Xin et al. (2020); Schuster et al. (2022); Xin et al. (2021); Tang et al. (2022) have achieved great success in improving efficiency. More concretely, Teerapittayanon et al. (2016) firstly appends a classifier after each convolutional neural network layer and utilizes entropy as a proxy to make exiting decisions. Xin et al. (2020) applies early exiting methods into BERT models and accelerates BERT with a slight performance drop. Tang et al. (2022) utilize early exiting strategies to accelerate vision language models and make early exiting decisions based on modalities and tasks. Moreover, Xin et al. (2021) proposes to learn to exit and extend their method into regression tasks. However, these models fail in diffusion models because of the time-series property of diffusion models in the reverse denoising process. Our method proposes uncertainty-aware layer-wise loss to preserve more information with fewer layers, achieving SoTA performance and efficiency trade-off compared with other methods.

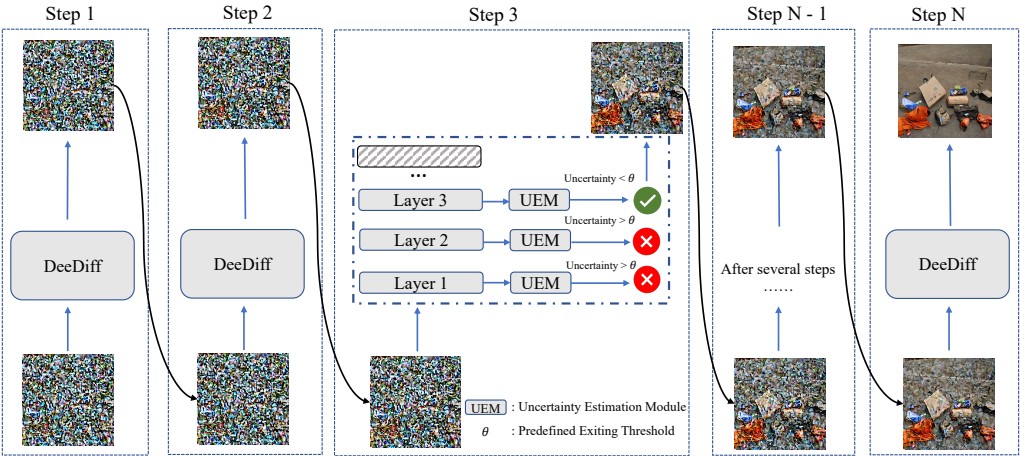

Figure 2: The overview of our proposed DeeDiff.

# 3 METHODS

## 3.1 PRELIMINARY

Diffusion models generally include two processes: a forward noising process and a reverse denoising process. The forward process is a Gaussian transition, which gradually adds noise to the original image following a predefined noise scheduler. The final image is expected to follow standard Gaussian distribution. The reverse process progressively removes the noise from data sampled from standard Gaussian distribution to generate high-fidelity images. The backbone network is utilized the estimated noise from noised images.

Formally, let $\mathbf{x}_0$ be the sample from training data distribution $q(\mathbf{x}_0)$, namely $x_0 \sim q(\mathbf{x}_0)$. A forward process adds noise gradually to $\mathbf{x}_0$ for $T$ times, generating a series of intermediate noisy samples $\{\mathbf{x}_1, ..., \mathbf{x}_T\}$ following:

$$q(\mathbf{x}_t|\mathbf{x}_{t-1}) = \mathcal{N}(\mathbf{x}_t; \sqrt{1\beta_t}\mathbf{x}_{t-1}, \beta_t\mathbf{I}) \tag{1}$$

$$\mathbf{x}_t = \alpha_t\mathbf{x}_0 + \beta_t\boldsymbol{\epsilon}. \tag{2}$$

where $\beta_t \in (0,1)$ is the variance scheduler and controls the level of the Gaussian noise added to data in each step, $\boldsymbol{\epsilon}$ refers to the noise sampled from standard Gaussian distribution $\mathcal{N}(0,\mathbf{I})$. Each forward step only depends on the previous step. Therefore, the forward process follows the Markov Chain property. Moreover, as long as $T \to \infty$, $\mathbf{x}_T$ approaches an isotropic Gaussian distribution.

The reverse process is formulated as another Gaussian transition, which removes noise in noisy images and restores the original images. However, since the reserve conditional distribution $q(\mathbf{x}_{t-1}|\mathbf{x}_t)$ is unknown at this point, diffusion models utilize neural networks to learn the real reverse conditional distribution $p_\theta(\mathbf{x}_{t-1}|\mathbf{x}_t)$.

$$p_\theta(\mathbf{x}_{t-1}|\mathbf{x}_t) = \mathcal{N}(\mathbf{x}_{t-1}; \tilde{\boldsymbol{\mu}}_{\theta,t}(\mathbf{x}_t), \tilde{\boldsymbol{\Sigma}}_{\theta,t}) \tag{3}$$

$\tilde{\boldsymbol{\mu}}_\theta$ and $\tilde{\boldsymbol{\Sigma}}_{\theta,t}(\mathbf{x}_t)$ are originally predicted statistics by backbone models. Ho et al. (2020) sets $\tilde{\Sigma}_{\theta,t}(\mathbf{x}_t)$ to the constant $\tilde{\beta}_t^2\mathbf{I}$, and $\tilde{\boldsymbol{\mu}}_\theta$ can be formulated as the linear combination of $\mathbf{x}_t$ and backbone noise estimation model $\boldsymbol{\epsilon}_\theta$:

$$\tilde{\mu}_{\theta,t}(\mathbf{x}_t) = \frac{1}{\sqrt{\alpha_t}}(\mathbf{x}_t - \frac{1-\alpha_t}{\sqrt{1-\bar{\alpha}_t}}\boldsymbol{\epsilon}_\theta) \tag{4}$$

$$\tilde{\beta}_t = \frac{1-\bar{\alpha}_{t-1}}{1-\bar{\alpha}_t} \cdot \beta_t \tag{5}$$

where $\alpha_t = 1 - \beta_t$, $\bar{\alpha}_t = \prod_{i=1}^{t}\alpha_i$. During training, Ho et al. Ho et al. (2020) found that using a simple loss function can achieve great performance:

$$\mathcal{L}_{\text{simple}}^t(\theta) = \mathbb{E}_{\mathbf{x}_0,\epsilon}\left[\|\epsilon - \boldsymbol{\epsilon}_\theta(\alpha_t\mathbf{x}_0 + \beta_t\epsilon)\|_2^2\right]. \tag{6}$$

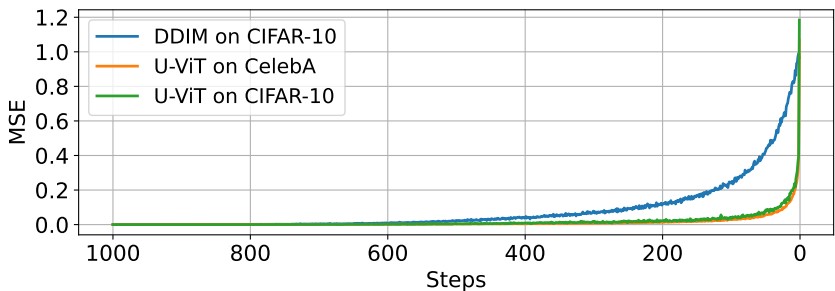

Figure 3: Average training loss distribution across 1000 steps on CIFAR-10 and CelebA.

## 3.2 Timestep-Aware Uncertainty Estimation Module

The basic assumption of early exiting is that the input samples of the test set are divided into easy and hard samples. The computation for easy samples terminates once some conditions are satisfied. Most early exiting methods attach a classifier following each intermediate layer to decide whether the network early exits or not. During inference, the output representations of each layer are fed into the classifier to obtain the confidence or entropy which is utilized as a proxy to represent the difficulty of samples and make early exiting decisions. We denote the backbone network consisting of $N$ layers, $L_i$ as the output representation of $i$-th layer, $i \in [1, N]$ and $CLS$ as the classifier.

$$C_{cls} = \max(\frac{1}{1 + e^{-CLS(L_i)}}) \text{ or } E_{cls} = \min(\sum_{c \in C} CLS(L_i)_c \log(CLS(L_i)_c)) \qquad (7)$$

If the $C_{cls}$ or $E_{cls}$ meets the predefined requirement such as the confidence is higher or the entropy is lower than the threshold, the computation terminates. Such a strategy works well in classification tasks. However, estimating the noise of each step of diffusion models can be considered as a regression task, which is challenging to estimate the confidence or entropy of predictions.

We consider each generation step separately and assume that the difficulty of the input varies for different generation steps. Hang et al. (2023) regards diffusion as multi-task learning and demonstrates that early-generation steps only require simple reconstructions of the input in order to achieve lower denoising loss. Specifically, we investigate the training loss distribution of each step which is able to reflect the difficulties of fitting to ground truth with different backbones and datasets, as shown in Figure 3. There is a clear distinction on the difficulty of different sampling steps, which demonstrates that the input of shallow and deep steps can be regarded as easy and hard samples. Inspired by Xin et al. (2021), we propose a lightweight timestep-aware uncertainty estimation module (UEM) to estimate the uncertainty of each prediction in diffusion models. Specifically, UEM is comprised of a fully-connected layer to estimate the uncertainty $u_i$ of each prediction:

$$u_{i,t} = f(\mathbf{w_t}^T[L_{i,t}, \text{ } timesteps] + b_t) \qquad (8)$$

where $\mathbf{w_t}, b_t, f, timesteps$ are the weight matrix, weight bias, activation function and timestep embeddings. We choose sigmoid as activation function since we hope the output range is $[0, 1]$. If the backbone is based on Transformer which requires tokenization, the output should be unpatched to generate uncertainty maps. The parameters are not shared between different uncertainty estimation modules since the module is able to learn the uncertainty with time-series properties for each layer. To introduce the supervision signal to this module, the pseudo uncertainty ground truth $\hat{u}_i$, which is negatively related to absolute error of the prediction and ground truth is constructed as:

$$\hat{u}_{i,t} = F(|g_i(L_{i,t}) - \epsilon_t|) \qquad (9)$$

where $g_i$ is the output layer and $\epsilon$ is the ground truth noise value. $F$ is a function to smooth the output of absolute error and we utilize $Tanh$ function in our experiments. The loss function of this module is designed as the MSE loss of estimated uncertainty and pseudo uncertainty ground truth:

$$\mathcal{L}_u^t = \sum_i^N \|u_{i,t} - \hat{u}_{i,t}\|^2 \qquad (10)$$

During inference, with the estimated uncertainty of the output prediction from each layer, the computation is able to make termination decisions once the uncertainty is lower than the predefined threshold. Moreover, since the generation process of diffusion models requires multiple inferences, we apply such an early exiting strategy to the inference each time, as shown in Figure 2.

### 3.3 UNCERTAINTY-AWARE LAYER-WISE LOSS

When a model makes an early exiting decision, it has to predict the output based on the current input and the information it has learned so far. Although it is confident enough for models to make early exiting decisions, there is always some information loss without full model Teerapittayanon et al. (2016). In order to address this issue and retain as much information as possible with fewer layers, previous works propose to utilize simple layer-wise loss for every intermediate layer:

$$\mathcal{L}_n^t = \sum_i^{N-1} \|g_i(L_{i,t}) - \epsilon\|^2 \tag{11}$$

This loss function is able to preserve information with fewer backbone layers and works well in many applications that only need inference once. However, such loss function fails in diffusion models, where multi-step inference is required. The main reason is that the information loss caused by early exiting accumulates as the sampling proceeds. The error accumulation figure is supplied in the Appendix. It is challenging for existing layer-wise loss to mitigate the accumulation of errors since they treat all timesteps equally. Hang et al. (2023) proved that finetuning specific steps benefits the surrounding steps in mitigating error accumulation. Inspired by this, we propose uncertainty-ware layer-wise loss:

$$\mathcal{L}_{UAL}^t = \sum_i^{N-1} (1 - u_{i,t}) \times \|g_i(L_{i,t}) - \epsilon\|^2 \tag{12}$$

where $u_i$ is the uncertainty value estimated in each layer. With such an uncertainty-aware loss function, on the one hand, timesteps and layers with low uncertainty represent the generation steps and layers that potentially contribute to mitigating the accumulation of errors. Higher weights on more important timesteps and layers are the key to filling the performance gap between full models and early-exited models. With the benefit of surrounding timesteps, more timesteps and layers tend to obtain higher certainty, which contributes to skipping more layers and achieving higher efficiency.

### 3.4 JOINT TRAINING STRATEGY

We have already discussed the uncertainty estimation loss in our early exiting framework and uncertainty-aware layer-wise loss for information preservation. There is an interdependence between these two loss functions. In order to balance the effect between uncertainty estimation loss and uncertainty-aware layer-wise loss, we utilize joint training strategy to optimize different loss functions simultaneously:

$$L_{all} = \mathcal{L}_{\text{simple}}^t(\theta) + \lambda \mathcal{L}_u^t + \beta \mathcal{L}_{UAL}^t \tag{13}$$

where $\lambda$ and $\beta$ are both hyper-parameters to balance simple task loss, uncertainty estimation loss and uncertainty-aware layer-wise loss. We set $\lambda$ and $\beta$ both to be 1 in our experiments.

## 4 EXPERIMENTS

### 4.1 EXPERIMENTAL SETUP

**Dataset.** CIFAR-10 Krizhevsky et al. (2009), Celeb-A Liu et al. (2015) are used to evaluate our methods and other methods on unconditional generation. ImageNet Deng et al. (2009) is a large-scale dataset with 1000 classes, which is used to evaluate class-conditional generation. Text-guided generation is evaluated on MS-COCO Lin et al. (2014) dataset. The resolution of ImageNet and COCO is both 256×256.

**Evaluation protocol.** We measure image generation quality utilizing Frechet Inception Distance (FID). Following Bao et al. (2022a), we compute FID with 50k generated samples on each task and dataset, with the reference generated from Dhariwal & Nichol (2021). As for efficiency, we report

Table 1: The performance and efficiency comparison between our method and other early exiting methods with the U-ViT-Small model on CIFAR-10, Celeb-A and MS-COCO dataset. Our method reduces computation compared with other methods while preserving well performance. w/o EE: without early exiting or the exiting threshold is 0. †: methods with 100 sampling steps.

| Methods | CIFAR-10 32×32 | | | CelebA 64×64 | | |
|---|---|---|---|---|---|---|
| | FID | Layers Ratio | GFLOPs | FID | Layers Ratio | GFLOPs |
| U-ViT | 3.11 | 1 | 22.86 | 2.87 | 1 | 23.01 |
| BERxiT | 20.5 | -20.5% | 18.18 (-20.4%) | 31.01 | -18.5% | 18.79 (-18.3%) |
| CALM | 20.0 | -20.5% | 18.18 (-20.4%) | 25.3 | -20.5% | 18.33 (-20.3%) |
| Static EE | - | - | - | 5.04 | -32.09% | - |
| Ours w/o EE | **2.7** | - | 22.86 | **2.63** | - | 23.01 |
| Ours | **3.7** | **-47.7%** | **11.97 (-47.6%)** | **3.9** | **-46.2%** | **12.48 (-45.7%)** |
| S-Pruning [†] | 5.29 | -44.2% | - | 6.24 | -44.3% | - |
| Ours [†] | 4.12 | -47.7% | - | 4.67 | -46.2% | - |

the number of average layers used in all test sample generation and its reduction ratio compared with baseline. Moreover, we provide the theoretical GFLOPs of each method on different datasets since the running time is unstable as it is influenced by hardware environments such as memory and I/O.

**Baseline.** Our methods are based on U-ViT Bao et al. (2022a), a transformer-based diffusion model. On ImageNet 256×256, we utilize the U-ViT-Large model with 21 layers while it is U-ViT-Small model with 13 layers on other datasets. We compare our method with existing early exiting methods, BERTxiT Xin et al. (2021) and CALM Schuster et al. (2022). Static early exiting methodMoon et al. (2023) uses a predefined exiting strategy to accelerate generation. S-Pruning Fang et al. (2023) is an accelerating method that applies structural pruning to diffusion models. Since the results reported in Fang et al. (2023) are based on 100 sampling steps, for a fair comparison, we also report our results with 100 steps. To show the generalizability of our method, we also conduct experiments on CNN-based diffusion models. The results are shown in the supplementary material.

**Implementation details.** During training, we utilize the AdamW Loshchilov & Hutter (2017) optimizer with a learning rate of 2e-4 for all datasets. For all tasks and datasets, we initialize our backbone with pre-trained weight from Bao et al. (2022a). The early exiting threshold are chosen from 0.2 to 0.01. More implementation details are shown in the supplementary material.

## 4.2 UNCONDITIONAL AND CLASS-CONDITIONAL GENERATION

We conduct unconditional generation experiments on CIFAR-10, Celeb-A and class-conditional generation on ImageNet 256 × 256. Following Bao et al. (2022a), we utilize DPM-Solver with 50 sampling steps in ImageNet and Euler-Maruyama SDE sampler with 1000 sampling steps in CIFAR-10 and Celeb-A datasets. The main results are shown in Table 1 and 2. Our methods achieve the best efficiency with minimal performance loss compared with other early exiting methods. More concretely, our method is able to reduce 47.7% layers on average and achieve 3.7 in FID while other early exiting methods such as BERTxiT and CALM only obtain layer reduction in around 20% and intense performance drop to 20 in FID on CIFAR-10 dataset. Moreover, on Celeb-A, the performance drop of BERTxiT and CALM is more severe with around 31 and 25 in FID and less than 20% layer number reduction while our method gains 3.9 in FID and uses 46.2% less layers. Similar results can also be found in large models with more layers on ImageNet. The FID of our method is 4.5 while the layer number is reduced to around 45.2%. In contrast, BERTixT and CALM incur severe performance drops to 23.5 and 21.4 respectively with less than 20% efficiency. The main reason is that our method helps the model to learn to use fewer layers to preserve more information and therefore achieve high efficiency with minimal performance loss. Interestingly, our model without early exiting strategy, achieves even better results than baseline models on CIFAR-10 and CelebA. More specifically, our pure model without early exiting obtains 2.7 FID value on CIFAR-10 and 2.63 on CelebA datasets, indicating that our novel uncertainty-ware layer-wise loss brings extra performance benefits for current Diffusion models.

Table 2: The performance and efficiency comparison between our method and other early exiting methods on ImageNet and COCO dataset.

| Methods | ImageNet 256×256 | | | MS-COCO 256×256 | | |
|---|---|---|---|---|---|---|
| | FID | Layers Ratio | GFLOPs | FID | Layers Ratio | GFLOPs |
| U-ViT | 3.4 | 1 | 142.20 | 5.95 | 1 | 25.04 |
| BERxiT | 23.5 | -17.8% | 116.89 (-17.7%) | 18.9 | -10.0% | 22.53 (-10.02%) |
| CALM | 21.4 | -19.3% | 114.76 (-19.2%) | 17.0 | -15.4% | 21.18 (-15.4%) |
| Ours w/o EE | 3.61 | - | 142.20 | 6.12 | - | 25.04 |
| Ours | **4.5** | **-45.2%** | **77.88 (-45.2%)** | **7.40** | **-43.6%** | **14.12 (43.6%)** |

## 4.3 TEXT-GUIDED GENERATION

In this section, we discuss the text-guided generation on MS-COCO dataset. Like on ImageNet, we utilize DPM-Solver with 50 sampling steps on MS-COCO. The main qualitative results are shown in Table 2. Our method achieves the best performance and efficiency trade-off compared with other methods. Specifically, DeeDiff obtains 7.4 FID with 43.6% computation reduction. In comparison, the FID of BERTxiT and CALM is 23.5 and 21.4 with 17.8% and 19.3% layer number reduction, respectively. The GFLOPs results are consistent with layer reduction ratio we report in the table.

## 4.4 ANALYSIS OF UEM AND UNCERTAINTY-AWARE LAYER-WISE LOSS

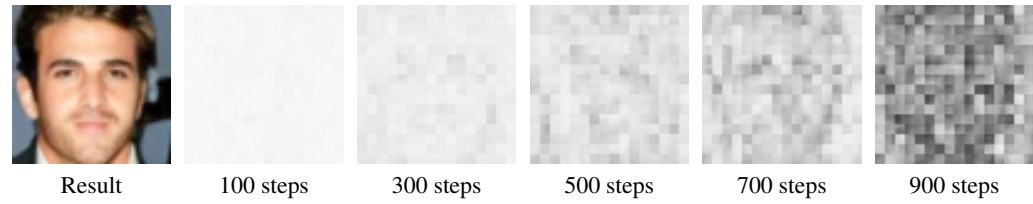

| Result | 100 steps | 300 steps | 500 steps | 700 steps | 900 steps |

Figure 4: The visualization of uncertainty map on different sampling steps of our DeeDiff.

In order to demonstrate the effectiveness of our proposed uncertainty-aware layer-wise loss, we provide a group of images that visualize the uncertainty map at different sampling steps as shown in Figure 4. At the beginning of the reverse denoising process, the uncertainty estimated by UEM module is insignificant since the noise estimation at the early stage is not accurate as well, which means shallow layers are able to generate similar results as deep layers at the early generation stage. However, as the input of the backbone includes less noise, the uncertainty value tends to increase. The larger the number of steps sampled, the higher the uncertainty of generating similar results as deep networks using early exited networks. The reason is that at the late stages of generation, more fine-grained noise estimation is required, which can be treated as hard examples for early exiting networks, resulting in more computation resource allocation.

## 4.5 PERFORMANCE AND EFFICIENCY TRADE-OFF

According to the discussion in Sec 4.2 and 4.3, our DeeDiff has achieved the best efficiency with minimal performance drop compared with other early exiting methods on all datasets. In this section, we discuss the performance and efficiency trade-off of our methods and other frameworks. We report the performance and efficiency trade-off curve of DeeDiff, BERTxiT and CALM on CIFAR-10, Celeb-A, and MS-COCO as shown in Figure 5. First of all, without doing early exiting, our method obtains the lowest FID value compared with BERTxiT and CALM. Moreover, BERTxiT and CALM tend to cause large performance drops with the increase of layer reduction ratio. In contrast, the trade-off curve of our DeeDiff is flat, which means our method is able to stay stable when more layers are skipped. Interestingly, at the highest efficiency point, our method achieves similar

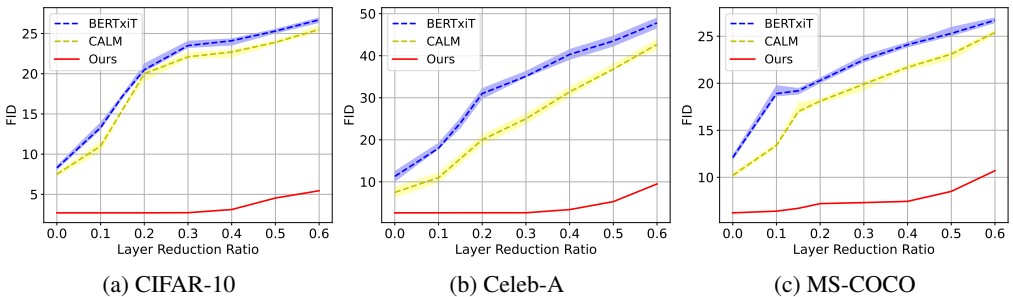

|  (a) CIFAR-10 | (b) Celeb-A | (c) MS-COCO |

Figure 5: Performance-Efficiency Trade-off comparison Curve on CIFAR-10, CelebA, COCO

performance to other methods such as BERTxiT and CALM at the lowest computation reduction point on CIFAR-10, which demonstrates the effectiveness of our method.

## 4.6 ABLATION STUDY

Table 3: Ablation study in unconditional image generation on CelebA. The results demonstrate the effectiveness of our proposed joint training strategy and uncertainty-aware layer-wise loss.

| Models | FID | Layers Ratio |
|---|---|---|
| DeeDiff w/o EE | 2.63 | - |
| DeeDiff | 3.9 | -46.2% |
| DeeDiff w/o UA-Loss | 14.5 | -20% |
| DeeDiff + Parameter Sharing | 5.7 | -41.7% |
| U-ViT-Small + DPM-Solver | 3.32 | - |
| DeeDiff + DPM-Solver w/o EE | 1.8 | - |
| DeeDiff + DPM-Solver | 3.9 | -42.3 % |

**DeeDiff without UA-Loss.** To analyze how uncertainty-aware layer-wise loss affects the whole model, we conduct a group of experiments that exclude uncertainty-aware layer-wise loss. As shown in the Table 3, the performance degrades to 14.5 with only 20% layer reduction, reflecting that our UA-loss is able to fill the performance gap between early-exited models and full models.

**Parameter Sharing.** We compare the model with parameter sharing and our models. The performance shown on the fourth line in Table 3 is 5.7 FID with 41.7% layer drop. We believe parameter sharing will be harmful to uncertainty estimation of diffusion models since the large sampling steps and number of layers make it hard for a simple linear layer to learn fine-grained uncertainty.

**Different Sampling Strategies.** We also conduct several experiments with DPM-Solver in 50 steps to show our model can bring benefits to other acceleration methods. Surprisingly, the model achieves even better results without early exiting compared with our basic models and shows similar performance as the basic model after early exiting. More discussion can be found in the appendix.

## 5 CONCLUSION & DISCUSSION

In this work, we propose a dynamic early exiting method to accelerate diffusion model generation. The uncertainty estimation module is aimed at estimating the uncertainty of the output. Our uncertainty-aware layer-wise loss concentrates on filling the performance gap. Our method achieves SoTA performance compared with other early exiting methods. However, there are limitations to our methods. First of all, although the performance and efficiency trade-off of our method is the best among other methods, DeeDiff still obtains high FID when the efficiency increases more than 60%. Second, our DeeDiff only considers the height of the generation process. The width of the generation process, namely the adaptive sampling steps is unexplored and left to future works.

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

# 6    APPENDIX

## 6.1    MORE EXPERIMENTAL SETUP

**Dataset & Baseline.** CIFAR-10 Krizhevsky et al. (2009), Celeb-A Liu et al. (2015) are used to evaluate our methods and other methods on unconditional generation. CIFAR-10 is a small-size dataset with a resolution of 32×32. The Celeb-A dataset, which includes 162,770 human faces, is a widely used dataset for unconditional image generation with a resolution of 64×64. ImageNet Deng et al. (2009) is a large-scale dataset with 1000 classes, which is used to evaluate class-conditional generation. Text-guided generation is evaluated on MS-COCO Lin et al. (2014) dataset. MS-COCO consists of 5 image captions for every image and contains 82,783 training images and 40,504 validation images. The resolution of ImageNet and COCO is both 256×256. We compare our method with existing early exiting methods, BERTxiT Xin et al. (2021) and CALM Schuster et al. (2022). Since BERTxiT and CALM are originally implemented on language model, we re-implement these methods on diffusion models. BERTxiT utilizes a learning strategy to extend early exiting to BERT models and apply average layer-wise loss ($L = \frac{1}{N}\sum_{i=1}^{N} L_i$, N is the number of layers) to train the network while CALM apply decay layer-wise loss ($L = \frac{i}{N}\sum_{i=1}^{N} L_i$). Furthermore, CALM uses the similarity of adjacent layers and confidence to decide to exit and calibrates local early exits from global constraints. In our experiments, we follow their training strategy and we only apply similarity to decide exiting for CALM since confidence-based exiting is hard to be applied to diffusion models. During training, we chose the best evaluation epoch of BERTxiT and CALM for a fair comparison.

**Implementation Details.** On ImageNet 256×256, we utilize the U-ViT-Large model with 21 layers and 16 attention heads while it is the U-ViT-Small model with 13 layers and 8 attention heads on other datasets. The hidden size of the network is 512 and 1024 for small and large models. We use a weight decay strategy and apply a weight decay of 0.03 for all datasets. We try the running coefficients $\beta_1, \beta_2$ of AdamW among 0.9, 0.99, 0.999, and find that ($\beta_1, \beta_2$) = (0.99, 0.99) performs well for all datasets. On ImageNet 256×256, ImageNet 512×512 and MS-COCO, we adopt classifier-free guidance Ho & Salimans (2022) following Rombach et al. (2022). We follow latent diffusion models (LDM) Rombach et al. (2022) for high-resolution image generation. Specifically, for images at 256×256 resolutions, we first convert the input images to latent representations at 32×32 resolutions, using the pre-trained image autoencoder provided by Stable Diffusion Rombach et al. (2022). All experiments are conducted on 4 A6000.

## 6.2    MORE EXPERIMENTAL RESULTS

### 6.2.1    STATISTICS OF EXITED LAYERS

We provide more experimental analysis about our proposed dynamic early exiting strategy. We report the average exited layer ratio of our model with different thresholds on the CIFAR-10 dataset, as shown in Figure 6. In the left figure, the experiment is conducted with a relatively low uncertainty threshold. Most computations are fully operated while there are still some parts of computation that are early stopped. In contrast, with a higher uncertainty threshold, most of the computations are terminated at the beginning which means some easy samples are allocated with less computation. This shows that with threshold control, our estimated uncertainty achieves a good trade-off between quality and efficiency.

### 6.2.2    RESULTS COMPARISON OF SMALL MODELS

In order to show the effectiveness of our method on top of the baseline model, we report the results comparison of our method with a small-size model that is trained from scratch. All training settings of the small model stay unchanged compared with baseline models except for layer number. The small model is comprised of 7 layers, which have about the same amount of computation as our model. Also, we choose the best evaluation epoch for a fair comparison. The results are shown in Table 4. Our method achieves better performance than the small model on CIFAR-10 and CelebA, which indicates the effectiveness of our method in accelerating the inference of diffusion models with minor performance reduction.

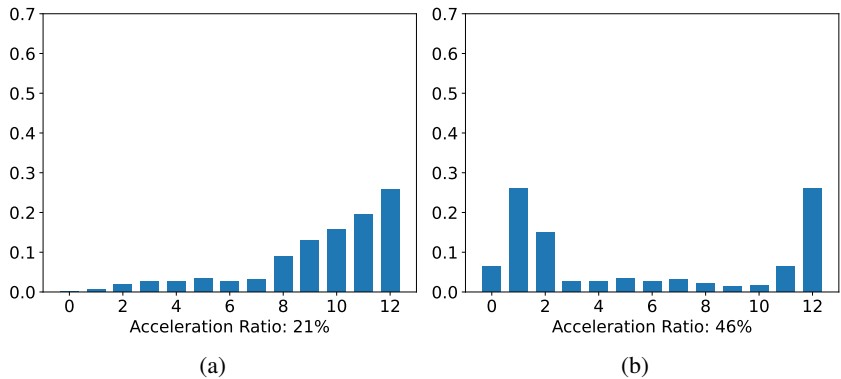

Figure 6: Statistics of exited layers on CIFAR-10

Table 4: Results comparison of small-size models and our method on CIFAR-10 and CelebA.

| Methods | CIFAR-10 | | CelebA | |
|---|---|---|---|---|
| | FID | GFLOPs | FID | GFLOPs |
| Small model | 6.68 | 12.8 | 4.58 | 13.0 |
| Ours | 3.7 | 11.97 | 3.9 | 12.48 |

### 6.2.3 RESULTS COMPARISON OF STATIC EXITING MODELS

To show the effectiveness of our dynamic early exiting method, we compare our method with the static exiting method, namely exits at a certain layer for all samples. The reported results are shown in Table 5. The static model is trained with our proposed UEM and uncertainty-aware layer-wise loss. All training settings are the same as our model. In the table, the model that exits at the 7th layer shares similar GFLOPs as our dynamic method. Our method achieves 3.7 and 3.9 FID on CIFAR-10 and CelebA while the static model only obtains 8.3 and 8.8 FID on each dataset, which demonstrates the effectiveness of our dynamic early exiting strategy.

Table 5: Results comparison of static model and our method on CIFAR-10 and CelebA.

| Models | CIFAR-10 | CelebA |
|---|---|---|
| Exit at 11th layer | 4.1 | 4.5 |
| Exit at 9th layer | 6.2 | 6.9 |
| Exit at 7th layer | 8.3 | 8.8 |
| Ours | 3.7 | 3.9 |

### 6.2.4 RESULTS ON CNN-BASED MODELS

Our method can be plugged into any framework easily such as Transformer and CNN. Therefore, we also provide the results on CNN-based models such as DDIM as shown in Figure 6. The backbone of DDIM applies the UNet structure, which consists of resolution upsampling and downsampling. To ensure resolution consistency, we force the output of each layer to be the same resolution by interpolation. We utilize the open-source code from the official repository and conduct a group of experiments on several datasets. Our experiments are conducted with different sampling steps, 50 and 100 steps. On CelebA and CIFAR-10, our DeeDiff is able to achieve around 40% layer reduction ratio with a slight performance drop.

Table 6: The performance and efficiency comparison of our DeeDiff on CNN-based models.

| Methods | CIFAR-10 | | CelebA | |
|---|---|---|---|---|
| | FID | Layers Ratio | FID | Layers Ratio |
| DDIM (50 steps) | 4.67 | 1 | 9.17 | 1 |
| Ours (50 steps) | 5.43 | -37.4% | 10.34 | -39.8% |
| DDIM (100 steps) | 4.16 | 1 | 6.53 | 1 |
| Ours (100 steps) | 4.98 | -40.8% | 7.35 | -41.5% |

Table 7: More results of different sampling strategies on CIFAR-10 and Celeb-A datasets. EM: Euler Maruyama sampling with 1000 steps. DPM: DPM-solver with 50 steps.

| Methods | CIFAR-10 32×32 | | CelebA 64×64 | |
|---|---|---|---|---|
| | FID | Layers Ratio | FID | Layers Ratio |
| U-ViT + EM | 3.11 | 1 | 2.87 | 1 |
| U-ViT + DPM | 4.5 | 1 | 3.3 | 1 |
| Ours + EM w/o EE | 2.7 | 1 | 2.63 | 1 |
| Ours + DPM w/o EE | 5.2 | 1 | 1.8 | 1 |
| Ours + EM | 3.7 | -47.7% | 3.9 | -46.2% |
| Ours + DPM | 6.9 | -43.1% | 3.7 | -41.3% |

### 6.2.5 RESULTS AND DISCUSSIONS ON DIFFERENT SAMPLING STRATEGIES.

In this section, we first provide more experimental results on our methods with different sampling strategies to show compatibility with other acceleration methods. Based on the results, we start an insightful discussion about the experimental observation and provide some assumptions for the results, which are left for future work to validate.

The results are shown in Table 7. There are several conclusions drawn from the results:

**The training of DeeDiff brings extra performance benefits.** Besides using DPM-Solver on CIFAR-10 datasets, our DeeDiff achieves better FID performance compared with baseline models. More specifically, utilizing EM sampling strategy with 1000 steps, U-ViT achieves 3.11 and 2.87 FID on CIFAR-10 and CelebA, respectively. Without acceleration via early exiting, DeeDiff obtains 2.7 and 2.63 FID on each dataset. With 50 sampling steps by DPM-solver strategy on CelebA, our method achieves surprisingly 1.8 FID while U-ViT only gains 3.3 FID, bringing more than 45% performance improvement.

**DeeDiff is compatible with other acceleration methods.** As shown in Table 7, DeeDiff is able to be combined with other acceleration methods. The performance keeps consistent with the combination of other methods. More concretely, utilizing Euler Maruyama sampling strategy, DeeDiff achieves 3.7 and 3.9 FID with 47.7% and 46.2% layer reduction on CIFAR-10 and CelebA respectively. Furthermore, for the DPM-Solver with 50 sampling steps, DeeDiff also obtain 6.9 and 3.7 FID on each datasets while the performance of baselines is 4.5 and 3.3 respectively.

**Discussion.** A natural question for the results in Table 7 is why the FID value of DeeDiff with DPM-solver is better than DeeDiff with Euler Maruyama. We believe the reason is the existence of model overfitting in pre-trained weights. With our UA-Loss, different training steps contribute unequally to model fitting, which potentially prevents the model from being overfitting. For future works, this could be a potential way to reduce the possibility for models to be overfitting.

### 6.3 ERROR CURVE COMPARISON

Besides uncertainty map visualization, we show the error accumulation curve of different methods in comparison with the baseline on several datasets, as shown in Figure 7. With early exited layers, the error of BERTxiT and CALM starts to increase as the number of sampling steps increments. The error curve of our method is always below that of other methods, which means that despite skipping

(a) CIFAR-10  (b) Celeb-A  (c) ImageNet  (d) MS-COCO

Figure 7: Error curve comparison of DeeDiff and other early exiting methods on CIFAR-10, Celeb-A, ImageNet and COCO.

many layers, the error accumulation of our method is still smaller than that of other methods. This proves that our proposed uncertainty-aware layer-wise loss is able to benefit the whole generation timesteps and thus incurs lower error accumulation.

## 6.4 MORE SAMPLES RESULTS

More generation samples can be found in Figure 8.

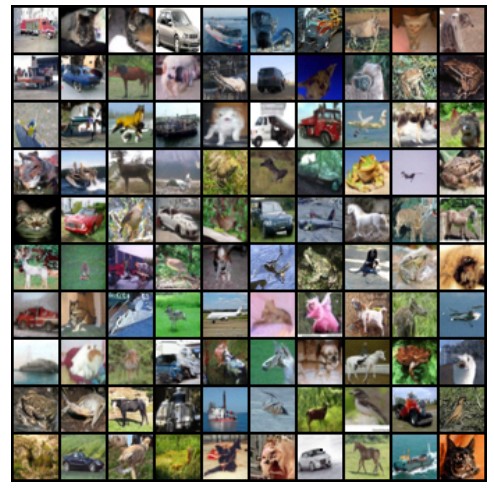

(a) CIFAR-10

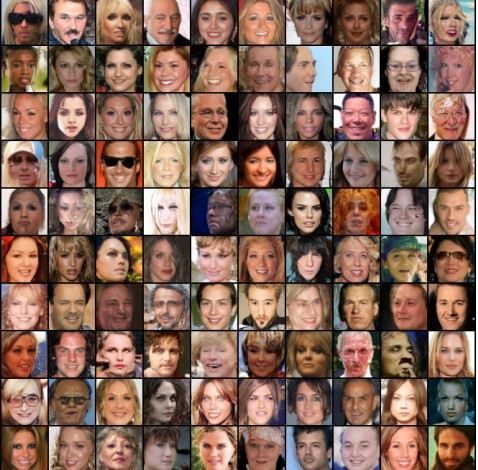

(b) Celeb-A

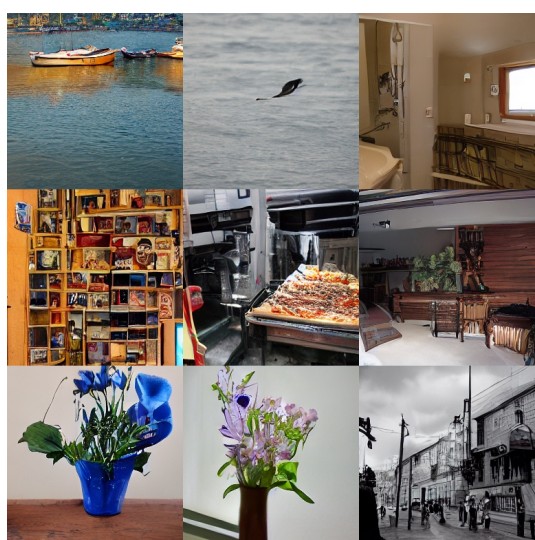

(c) MS-COCO

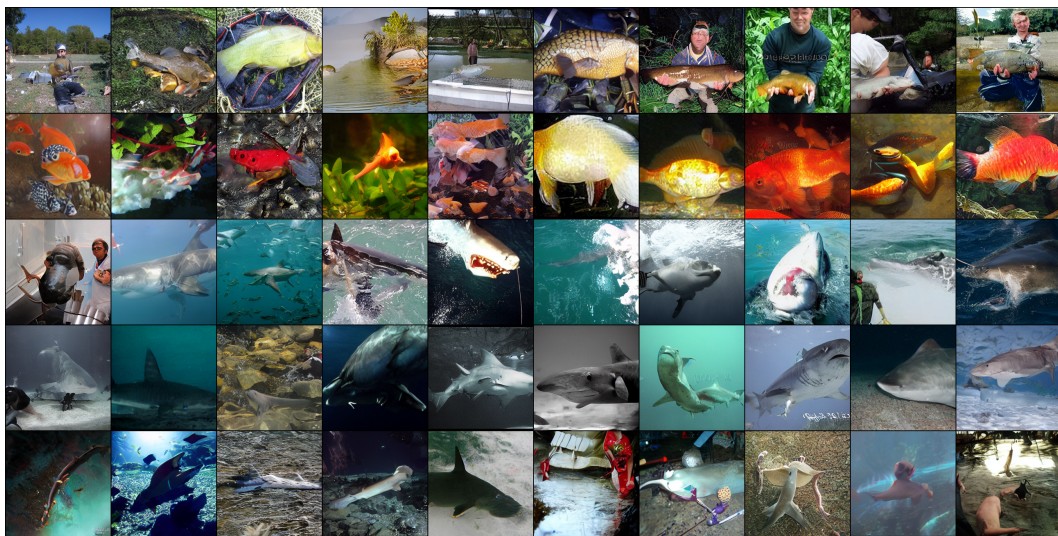

(d) ImageNet

Figure 8: Generation Samples on CIFAR-10, Celeb-A, MS-COCO and ImageNet

