# OpenReview forum: "DeeDiff: Dynamic Uncertainty-Aware Early Exiting for Accelerating Diffusion Model Generation"
_ICLR.cc/2024/Conference — Submitted to ICLR 2024_

### Official Review · Reviewer_ttQR · 2023-10-27

**Soundness:** 3 good
**Presentation:** 2 fair
**Contribution:** 2 fair
**Rating:** 5
**Confidence:** 4

**Summary:**

This study presents an early-exit method designed to speed up the inference process in diffusion probability models. At the heart of this approach is an uncertainty estimation module, which quantifies uncertainty in terms of prediction error. The model is designed to terminate the inference process at the layer once the uncertainty surpasses a pre-set threshold. To enhance performance and minimize the discrepancy between predictions made through early exit and those using the full model, the paper introduces an uncertainty-aware, layer-wise loss for training the diffusion model. Empirical evaluations conducted on datasets such as CIFAR-10, CelebA, ImageNet-256, and MS-COCO-256 have yielded encouraging results.

**Strengths:**

1. The concept of an early exit strategy presents a practical and effective approach to accelerating diffusion models. This idea is notably complementary to other acceleration techniques, such as efficient sampling.
2. The paper thoroughly investigates its hypotheses through extensive experiments on several well-known benchmarks, including ImageNet and COCO.
3. The manuscript is commendably well-written, featuring clear illustrations and well-defined formulations.

**Weaknesses:**

This is a solid work, but I still have some concerns about the novelty and fairness.

1. There is notable prior research in the area of early exiting within diffusion models, such as the study presented at the ICML-23 workshop [1]. This context suggests that the novelty of the current paper might be somewhat constrained, though it undoubtedly contributes to the ongoing discourse in the field.
2. Regarding the comparisons made in Table 1, I would like to express some reservations about their fairness. For instance, the adaptation methods of BERxiT and CALM for diffusion models aren't entirely clear. Also, there are some setting difference between the proposed method and S-Pruning [2], which uses a 100-step DDIM and a smaller UNet (6.1G MACs). But the proposed method employs a 1000-step Euler-Maruyama SDE sampler with a larger network (11.97 GFLOPS). Besides, It would be greatly appreciated if the authors could clarify whether the GFLOPS mentioned are synonymous with Multiply-Accumulate Operations (MACs). A detailed explanation of how GFLOPS are calculated would be helpful, particularly since many popular libraries, such as PyTorch-opcounter [3] computes MACs by default.
3. The paper introduces an uncertainty-aware layer-wise loss, enhancing the DDPM objective by prioritizing steps with small uncertainty. However, given that diffusion models typically show lower prediction errors in earlier steps, as illustrated in Figure 3, does this mean that the proposed loss method just simply focuses more on these initial steps? Also, I'd like to gently point out a possible minor error in Figure 3, where step 0 is actually the final step [4] rather than initial step.
4. The citation format can be improved. There are several citation issues such as "Ho et al. Ho et al. (2020)" (Bellow Eqn 3),  " S-PruningFang et al. (2023)" (The baseline subsection in 4.1).

[1] Moon, Taehong, et al. "Early Exiting for Accelerated Inference in Diffusion Models." ICML 2023 Workshop on Structured Probabilistic Inference {\&} Generative Modeling. 2023.
[2] Fang, Gongfan, Xinyin Ma, and Xinchao Wang. "Structural Pruning for Diffusion Models." arXiv preprint arXiv:2305.10924 (2023).
[3] Ligeng Zhu, “PyTorch-opcounter”, GitHub repository.
[4] Ho, Jonathan, Ajay Jain, and Pieter Abbeel. "Denoising diffusion probabilistic models." Advances in neural information processing systems 33 (2020): 6840-6851.

**Questions:**

Please refer to the weaknesses.

---

> ### Author Response · Authors · 2023-11-20
> **Response to Reviewer ttQR**
>
> Thank you very much for your detailed and constructive review.
>
> ***Response to weakness 1***:
>
> Thank you for providing more related recent work on our topic. We will cite this work in our revised version.
>
> Indeed, the ICML-23 workshop[1] paper shares similar ideas with us. However, there are a few key points that set us apart from them. First of all, the exiting strategy in [1] is static and pre-defined by hand and the model requires finetune under each strategy, which in fact introduces more cost and instability to achieve acceleration. Instead, our DeeDiff learns dynamic exiting strategy and the performance and efficiency trade-off are simply controlled by the threshold. Second, our method achieves better performance with higher efficiency compared with [1]. For example, [1] obtains 4.98 FID with 32.09\% acceleration ratio while our method gains 3.9 FID with 46.2\% acceleration ratio.
>
> ***Response to weakness 2***:
>
> We apologize for the lack of detailed descriptions. We will provide more details for BERTxiT and CALM in the revised version. In short, BERTxiT utilizes a learning strategy to extend early exiting to BERT models and apply average layer-wise loss ($L = \frac{1}{N} \sum_{i=1}^N L_i$, N is the number of layers) to train the network while CALM apply decay layer-wise loss ($L = \frac{i}{N} \sum_{i=1}^N L_i$). Furthermore, CALM uses the similarity of adjacent layers and confidence to decide to exit and calibrates local early exits from global constraints. In our experiments, we follow their training strategy and we only apply similarity to decide exiting for CALM since confidence-based exiting is hard to be applied to diffusion models. During training, we chose the best evaluation epoch of BERTxiT and CALM for a fair comparison.
>
> As for the comparison with S-Pruning[2], the GFLOPs we reported are not synonymous with Multiply-Accumulate Operations (MACs). Typically, 1 MACs roughly equals 2 GFLOPs which means that our backbone has slightly less computation cost than S-Pruning (11.97 vs 12.2). We apologize for our fault in the sampling step. We would like to provide the performance of our method on DDIM 100 sampling steps as follows:
>
> | Methods            | CIFAR-10 | CelebA |
> |--------------------|----------|--------|
> | S-Pruning          | 5.29     | 6.24   |
> | Ours with 100 step | 4.12     | 4.67   |
>
> ***Response to weakness 3***:
>
> The answer is partly yes. At the beginning of training, the loss will focus more on early stages. However, with further optimization, the loss will gradually focus on the latter steps, achieving globally optimal.
>
> Thank you very much for pointing out our minor mistake. We will fix it in our revised version.
>
> ***Response to weakness 4***:
>
> We apologize for the faults of citation and thank reviewer for your detailed comments. We will update the citation in the revised version.
>
> [1] Moon, Taehong, et al. "Early Exiting for Accelerated Inference in Diffusion Models." ICML 2023 Workshop on Structured Probabilistic Inference & Generative Modeling. 2023.
>
> [2] Fang, Gongfan, Xinyin Ma, and Xinchao Wang. "Structural Pruning for Diffusion Models." arXiv preprint arXiv:2305.10924 (2023).

---

> > ### Comment · Reviewer_ttQR · 2023-11-23
> > **Response**
> >
> > Dear authors,
> >
> > Thanks for providing more details. Most of the weaknesses have been addressed. So, I will adjust my rating.

---

> > > ### Author Response · Authors · 2023-11-23
> > > **Response to Reviewer**
> > >
> > > Dear reviewer,
> > >
> > > Thank you very much for your reply. We truly appreciate your suggestions for our work. We wish you a good day!

---

### Official Review · Reviewer_xWTy · 2023-10-29

**Soundness:** 3 good
**Presentation:** 1 poor
**Contribution:** 2 fair
**Rating:** 3
**Confidence:** 4

**Summary:**

The paper proposes DeeDiff to accelerate diffusion model generation. Specifically, DeeDiff employs an early exiting strategy where the output can be directly derived from the early layers at different timesteps based on the uncertainty estimation module (UEM). Experiments are conducted on CIFAR, ImageNet, and MS-COCO with FID score to show its effectiveness.

**Strengths:**

1. The paper makes an pioneering investigation of early exiting with Diffusion Model which is of novelty.
2. The proposed method is effective with reported FID closer to the full-size model at around 40% FLOPs reduction, outperforming the other early exiting methods.

**Weaknesses:**

1. The experiment results are not convincing.
    - the paper claims that their method reduces the inference time by up to 40%, while the results section only presents the FLOPs reduction. It is obvious that run-time speedup can not be directly represented by the theoretical FLOPs reduction, and thus the claim is falseful. In fact, with these overheads, it is hard to know how much actual speedup this method can bring.
    - the most noticeable capability of a diffusion model, text-guided generation, is not well evaluated. Only FID score is shown, while not a single visual figure is shown. Also, the image-text alignment is not evaluated which is another widely-used metric to assess diffusion model's quality.
2. The presentation lacks clarity.
    - in Figure 4, I have 0 idea what it is about. What does the level of grayness mean? Where are the uncertainty maps from? It makes me hard to understand the analysis.
    - the methodology presented in Section 3.2 is also not clear to me. Since there are 2 dependent indices, $t$ and $i$, it is worthwhile to mention the dependency for the matrices of $w_t$, $b_t$, and $g_i$. Specifically, is $g_i$ first learned and then fixed afterwards for learning $w_t$ and $b_t$? It looks to me $g_i$ shall be fixed first to ensure a low $\hat{u}_{i,t}$ or otherwise the learning seems incorrect to me. It would be good to present a flow-chart/figure to understand the learning scheme for these parameters as well.

**Questions:**

Please see the weakness section.

---

> ### Author Response · Authors · 2023-11-20
> **Response to Reviewer xWTy**
>
> Thank you very much for your detailed and constructive review.
>
> ***Response to weakness 1***:
>
> The running time is unstable since it is influenced by hardware environments such as memory and I/O. Therefore, we provide GFLOPs to show the theoretical computation cost for fair comparison. The extra computation cost inducted by the UEM is negligible compared with the acceleration brought by our methods.
>
> We apologize for the lack of text-to-image generated results and text-image alignment. We will add images and alignment metrics in the revised version.
>
> ***Response to weakness 2***:
>
> We apologize for the unclearness of our notations. The grayness refers to the uncertainty value, where the grayer points mean higher uncertainty. The uncertainty map is generated from UEM. During inference, we will compute the average uncertainty at the image level and utilize it as a proxy to make exiting decisions.
>
> In notations, $g_i$ is in fact the output layer used in U-ViT to reshape transformer tokens back to images, which are the final output for generation. $w_t$ and $b_t$ are the main learned parameters of UEM. Therefore, $g_i$ is not fixed during training. Also, the training objective is to minimize the MSE of $u_{i,t}$ and $\hat{u}_{i,t}$ since we hope to optimize $w_t$ and $b_t$.

---

### Official Review · Reviewer_eCAn · 2023-11-03

**Soundness:** 3 good
**Presentation:** 3 good
**Contribution:** 2 fair
**Rating:** 5
**Confidence:** 4

**Summary:**

This paper proposes a early exiting framework for accelerating diffusion models.  The authors propose a timestep-aware uncertainty estimation module given the multistep sampling, and an uncertainty-ware layer-wise to fill the performance gap.

**Strengths:**

The authors extend the early exiting approaches to the diffusion models, which considers the feature of multistep sampling of diffusion models. The proposed method shows promising results in accelerating diffusion models.

**Weaknesses:**

1.	The training cost. It seems the costs brought by the UEM loss and the layer-wise loss are high. It seems the method needs to backward at each layer. It’d be better to clarify the extra costs brought by the proposed method. Besides, discusssion about the scalability might also be important, i.e., is the method suitable for large diffusion models such as stable diffusion.
2.	Maybe the comparison to some heuristic settings is needed to demonstrate the effectiveness of the proposed aumoted exiting mechanism. For example, exit at a fixed layer for all inputs.

**Questions:**

Please see the weaknesses.

---

> ### Author Response · Authors · 2023-11-20
> **Response to the Weakness of Reviewer eCAn**
>
> Thank you a lot for your detailed and constructive review.
>
> ***Response to weakness 1***:
>
> Our method does not significantly increase the training cost. On the one hand, UEM is a basic linear layer that entails minimal computation costs and few parameters. On the other hand, our model utilizes the pretrained weight upon the baseline model which brings benefits to model convergence. We will provide the exact training cost compared with other methods in the revised version. Furthermore, the cost added in inference is negligible compared with the acceleration brought by our methods. Therefore, our method is also able to transfer to large diffusion models such as stable diffusion.
>
> ***Response to weakness 2***:
>
> Thank you very much for your detailed advice. We provide the performance of the models that exit with fixed layers as follows:
>
> | Methods          | CIFAR-10 | CelebA |
> |------------------|----------|--------|
> | Exit at 11 layer | 4.1      | 4.5    |
> | Exit at 9 layer  | 6.2      | 6.9    |
> | Exit at 7 layer  | 8.3      | 8.8    |
> | Ours             | 3.7      | 3.9    |
>
> In the table, 'Exit at 7 layer' shares similar GFLOPs with our dynamic method. The table shows that our dynamic method gains better performance than the static exiting method, demonstrating the effectiveness of our proposed dynamic exiting strategy.

---

### Official Review · Reviewer_PB4D · 2023-11-04

**Soundness:** 2 fair
**Presentation:** 2 fair
**Contribution:** 2 fair
**Rating:** 5
**Confidence:** 4

**Summary:**

Taking the spirit of early existing techniques in transformers, the paper proposes to extend it to diffusion models. To enable early existing, the paper introduces an uncertainty estimation module (UEM) to characterize the sampling uncertainty at each layer, and an uncertainty-weighted loss to better integrate the UEM in the network. Empirically the proposed DeeDiff framework improves the sample efficiency across datasets.

**Strengths:**

- The paper introduces an uncertainty estimation module and layer-wise loss in order to enable the early existing in diffusion models.

- Experimentally, the proposed method improved over the baseline across datasets in terms of sample efficiency.

**Weaknesses:**

- Learning the targeted error $\hat{u}_{i,t}$ in Eq.10 appears to be a very challenging task. I am skeptical that the simplistic one-layer neural network presented in Eq.8 cannot capture the per-sample uncertainty. My guess is that the UEM may only learn a sample-independent value. In this case, the UEM module is effectively a simple pruning technique. Could the authors provide a more empirical analysis of the UEM module?

- (continuing on the point above) I'm not surprised that the model trained with the layer-wise loss (Ours w/o EE) can provide better performance if the UEM module is actually doing the simple pruning. On these simple datasets considered in this paper (CIFAR-10, CelebA), people often observe performance gain after shrinking the architecture. For example, in EDM [1] Table 7, reduce the number of layers in the original config. B to config. C-F improves the performance.

- I don't think "Ours w/o EE" can improve over the baseline in more complicated datasets like ImageNet. Could you also report "Ours w/o EE" on ImageNet-256 and MS-COCO-256? I would imagine the layer-wise loss ("Ours w/o EE") could hurt the overall performance when the network capacity falls short.

- Could the authors provide some description of the BERTxiT and CALM, as well as how they are applied to diffusion models?

- Is the proposed approach limited to transformer architecture? It seems that the proposed method is only applicable to architecture with a constant feature dimensionality. The more popular UNet architecture has a varying feature dimensionality.

- The notation is a bit unclear: Could you clarify what $g_i$ and $L_{i,t}$ are? To my understanding. $L_{i,t}$ is the output features of the $i$-th layer. It's unclear to me what's the operator $g_i$ on top of $L_{i,t}$.

- Overall, the reviewer feels like the proposed method is simply a layer-wise pruning method, with the error $u_{i,t}$ as the guidance. One simple baseline is to retrain a **smaller** diffusion model from scratch, that uses similar GFLOPs with "Ours" in Table 1, and see how it performs.

[1] Karras et al, Elucidating the Design Space of Diffusion-Based Generative Models, NeurIPS 22.

**Questions:**

- Could the authors provide more details for Fig 1? Could you clarify which time's MSE you are reporting? Is this 13-layer Transformer trained with the proposed Layer-wise loss?

- From Fig. 4, it seems that the uncertainty map $u_{i,t}$ is a feature map rather than a real number?

- Is the $u_{i,t}$ fixed in Eq.12? (the training in Eq.10 finishes before Eq.12).

---

> ### Author Response · Authors · 2023-11-20
> **Response to the Weakness of Reviewer PB4D**
>
> Thank you very much for your detailed and constructive review.
>
> **Response to weakness 1**:
>
> UEM should not be dismissed as a mere pruning technique. UEM is able to learn the uncertainty per sample. We sincerely invite reviewers to refer to Figure 4. The gray images are generated by UEM, where the grayer points mean higher uncertainty. It shows that UEM can learn unique features for each sample such as the face structure shown in Figure 4. Moreover, in order to distinguish our method from simple pruning, we will provide the statistics of samples using different average layers in our revised version (Please check the appendix).
>
> **Response to weakness 2 and 3**:
>
> We would like to respectfully clarify the reviewer's misunderstanding of UEM and the uncertainty-aware layer-wise loss. Our proposed UEM and loss do not alter the structure of the base model during training. The experiments of ours w/o EE follow the same experimental settings (number of layers) as the baseline model. Furthermore,  previous works that apply simple layer-wise loss generally meet performance drops after training because of hard optimization. For example, in MuE[1], please compare the results of the first line in Table 1 with the first line in Table 2. The performance drops slightly with simple layer-wise loss while our proposed uncertainty-aware layer-wise loss brings extra benefits to performance on CIFAR-10 and CelebA. We also report the ours w/o EE performance on COCO and ImageNet.
>
> | Models        | ImageNet FID | COCO FID |
> |---------------|--------------|----------|
> | BERxiT w/o EE | 7.32         | 9.08     |
> | CALM w/o EE   | 6.84         | 8.56     |
> | Ours w/o EE   | 3.61         | 6.12     |
>
> Our method achieves superior performance compared to other early-exiting techniques, without exiting during inference. We believe that the extent of additional benefits our method can provide is dependent on the size of the datasets.
>
> **Summary response to weakness 1, 2, 3**:
>
> While we are happy to further discuss the added benefits of our method, we would like to emphasize its primary contribution - that DeeDiff achieves the best efficiency while minimizing performance loss. The additional benefits provide us with a deeper understanding of diffusion models, which we will discuss more in the revised version.
>
> **Response to weakness 4**:
>
> We apologize for the lack of detailed descriptions. We will provide more details for BERTxiT and CALM in the revised version. In short, BERTxiT utilizes a learning strategy to extend early exiting to BERT models and apply average layer-wise loss ($L = \frac{1}{N} \sum_{i=1}^N L_i$, N is the number of layers) to train the network while CALM apply decay layer-wise loss ($L = \frac{i}{N} \sum_{i=1}^N L_i$). Furthermore, CALM uses the similarity of adjacent layers and confidence to decide to exit and  calibrates local early exits from global constraints. In our experiments, we follow their training strategy and we only apply similarity to decide exiting for CALM since confidence-based exiting is hard to be applied to diffusion models. During training, we chose the best evaluation epoch of BERTxiT and CALM for a fair comparison.
>
> **Response to weakness 5**:
>
> Our method is not limited to Transformer architecture. We would like to reviewer to check Table 4 in the appendix. We provide the performance and efficiency of our method applied in CNN-based diffusion models.
>
> **Response to weakness 6**:
>
> We apologize for the lack of clarity for notations. $L_{i,t}$ is the output features of the $i$-th layer at $t$ timestep. In U-ViT[2], there is an output layer that maps the tokens to noise images, namely $g_i$. We would like to invite the reviewer to refer to figure 1 in U-ViT[2].
>
> **Response to weakness 7**:
>
> Thank you for your generous suggestion. We report the performance of the small model here:
> | Methods     | CIFAR-10 FID | GFLOPs | CelebA FID | GFLOPs |
> |-------------|--------------|--------|------------|--------|
> | Small Model | 6.68         | 12.8   | 4.58       | 13.0   |
> | Ours        | 3.7          | 11.97  | 3.9        | 12.48  |
>
> The small model is trained with 7 layers from scratch. All settings stay unchanged compared with baseline models. Also, we choose the best evaluation epoch for a fair comparison.
>
>
> [1] Tang S, Wang Y, Kong Z, et al. You Need Multiple Exiting: Dynamic Early Exiting for Accelerating Unified Vision Language Model[C]//Proceedings of the IEEE/CVF Conference on Computer Vision and Pattern Recognition. 2023: 10781-10791.
>
> [2] Bao F, Nie S, Xue K, et al. All are worth words: A vit backbone for diffusion models[C]//Proceedings of the IEEE/CVF Conference on Computer Vision and Pattern Recognition. 2023: 22669-22679.

---

> > ### Author Response · Authors · 2023-11-20
> > **Response to the Questions of Reviewer PB4D**
> >
> > ***Response to Question 1***:
> >
> > The MSE is the average value of all timesteps. On CIFAR-10 and CelebA, the baseline model utilizes smaller models with 13 layers while on ImageNet, the model obtains 21 layers. The conclusion of the figure is that the network used for each dataset includes redundancy. Moreover, the 13-layer Transformer is not trained with our proposed layer-wise loss. The results come from the baseline model, U-ViT instead of being trained by layer-wise loss.
> >
> > ***Response to Question 2***:
> >
> > Yes. The UEM first generates the uncertainty map. During inference, we compute the average uncertainty across the image and obtain the average uncertainty as a metric to make an exiting decision.
> >
> > ***Response to Question 3***:
> >
> > Yes. Eq. 10 and 12 are combined in Eq.13 to optimize the objective together. During training, we utilize the 'detach' operator on $u_{i,t}$ for stable training.

---

### Author Response · Authors · 2023-11-20
**To All Reviewers**

Thank you very much for all your detailed reviews. Your comments are the important sources to improve our work. We have already replied to every constructive comment. If you still have further questions, please do not hesitate to write your response to us. We will reply as soon as possible before the end of discussion period. Later on, we will upload the revised version of our paper.

Thank you all again for your support and suggestion to our work. We wish you a good day and happy life.

---

> ### Author Response · Authors · 2023-11-21
> **Revised Version is Available**
>
> Dear reviewers and ACs,
>
> The revised version is available now. Thank you all for your support and review. The updated contents are marked as red words. Since the page limit, most updated experiments are added in appendix. We sincerely invite all reviewers to check our revised version. We hope to have more discussions with reviewers. Thanks again!

---

### Meta-Review · Area_Chair_pjYT · 2023-12-05

**Metareview:**

The paper extends the early exiting approaches to the diffusion models by estimating the uncertainty of the output in each layer. The method can reduce the GFLOPs compared to diffusion (e.g., U-ViT) and other exiting methods given 1000 steps or 100 steps.

According to the reviewers, the significance of the paper is limited. For instance, users may care about the clock time more than GFLOPs and it is unclear how the proposed method works in a typical setting: e.g., a fast sampler with 20 steps.

Although the authors provided a rebuttal, all reviewers voted for rejection.

**Justification For Why Not Higher Score:**

Although the authors provided a rebuttal, all reviewers voted for rejection.

**Justification For Why Not Lower Score:**

N/A

---

### Decision · Program_Chairs · 2024-01-16

Reject